# Malignant neoplasms in people with hypothyroidism in Spain: A population-based analysis

**Juan J. Díez**[1,2]*, **Pedro Iglesias**[1,2]

**1** Department of Endocrinology, Instituto de Investigación Sanitaria Puerta de Hierro Segovia de Arana, Hospital Universitario Puerta de Hierro Majadahonda, Majadahonda, Spain, **2** Department of Medicine, Universidad Autónoma de Madrid, Madrid, Spain

\* juanjose.diez@salud.madrid.org

**Data Availability Statement:** All data underlying the results described in this paper are fully available and can be found in the Supporting information files.

## Abstract

### Purpose

The objective of this study was to determine the association between hypothyroidism and overall and site-specific cancer in Spanish population.

### Methods

A cross-sectional study was performed using the population-based database BDCAP (Base de Datos Clínicos de Atención Primaria, primary care clinical database) to analyze the relative risk of cancer in Spanish population with hypothyroidism.

### Results

In a total of 2,414,165 patients diagnosed with hypothyroidism in BDCAP in 2019, the relative risk (OR) of cancer, compared to the non-hypothyroid population, was 1.73 (1.72–1.74) (P<0.0001). The higher risk was observed in both men (OR 2.15 [2.13–2.17]; P<0.0001) and women (OR 1.67 [1.636–1.68]; P<0.0001). However, hypothyroid persons aged 65 years or older had a reduced risk of cancer (OR 0.98 [0.97–0.98]; P<0.0001). In addition, hypothyroid patients aged 65 or over showed a decreased risk of cancers of the bladder, colorectal, gastric, pancreatic and prostate. Socioeconomic characteristics such as income level, municipality size, country of birth and employment situation had limited influence on the association between hypothyroidism and cancer. However, hypothyroid patients receiving replacement therapy exhibited higher cancer risk compared with patients without treatment (OR 1.30 [1.28–1.31]; P<0.0001).

### Conclusion

Spanish hypothyroid patients of both genders have a risk of overall cancer higher than that found in non-hypothyroid population. However, people over 65 years have a reduced risk of various malignancies. This peculiarity of hypothyroidism should be considered by the health authorities.

**Funding:** The authors received no specific funding for this work.

**Competing interests:** The authors have declared that no competing interests exist.

## Introduction

Cancer is one of the most common causes of morbidity and mortality worldwide. The International Agency for Research on Cancer estimated that approximately 18.1 million new cases were diagnosed in the world in 2020 [1]. In Spain, approximately 280,000 malignant tumors are diagnosed each year and cancer is the first cause of mortality in men and the second in women [2]. Hypothyroidism is one of the most prevalent endocrine disorders in clinical practice and its prevalence in the Spanish population has been estimated at 8.8% [3]. It is well known that hypothyroidism is frequency associated to comorbidities, such as diabetes, hyperlipidemia, and cardiovascular disease [4].

Experimental studies have suggested a relationship between thyroid function and cancer development [5]. Triiodothyronine (T3) can interact with nuclear thyroid hormone receptors (TR) to modulate transcriptional activities via thyroid hormone response elements in the regulatory regions of target genes or bind receptor molecules showing no structural homology to TRs, such as the cell surface receptor site on integrin αvβ3. Additionally, thyroxine (T4) binding to integrin αvβ3 is reported to induce gene expression through initiating non-genomic actions, further influencing angiogenesis and cell proliferation [6]. Thyroid hormones have been reported to impact cancer development and growth [7, 8], but also the deficiency of these hormones has been related to an elevated risk of cancer [9] and a worse prognosis of some malignancies [10–12]. In addition, epidemiological studies further supported these preclinical observations. High T4 levels have been reported to be associated with an increased risk of any solid, lung, and breast cancer [13]. Untreated hypothyroidism has been related to higher risk of common cancers such as breast [14, 15] and colorectal cancer [16].

Despite promising evidence from these preclinical and epidemiological investigations, the relationships between cancer and thyroid dysfunction are still poorly understood [17]. Comprehensive analyzes in large population groups of the association between thyroid hormone deficiency and common cancer types may shed light on these issues. The Base de Datos Clínicos de Atención Primaria (BDCAP, Primary Care Clinical Database) of the Spanish National Health System collects codified and standardized clinical information on an annual basis on the care provided at the primary level of care. The data are extracted from the medical records of the population assigned to primary care throughout all the country. The included variables encompass active health problems, interventions performed, and a selection of intermediate health outcomes [18]. The main objective of this study has been the description, in patients with hypothyroidism, of the relative risk of overall cancer and the most common neoplasms in the Spanish population, using the information available in the BDCAP database. A secondary objective was to analyze whether the relative risks found are modified by the demographic and socioeconomic characteristics of the subjects, as well as by thyroid hormone replacement therapy.

## Material and methods

### Study design

We performed an observational, retrospective, non-interventional study using the statistical portal of the Ministry of Health associated with the BDCAP database [19]. This database has been built for statistical and research purposes, with the consensus of all the autonomous communities of Spain. It annually collects standardized clinical information on the care provided by the Primary Care level and is representative of each autonomous community. This study was approved by the local ethics committee of the Hospital Universitario Puerta de Hierro Majadahonda (Madrid, Spain) (PI 94/22). Since our study was carried out through a database with accumulated information, the need for consent was waived by the ethics committee.

The latest update of this database in 2019 contains information from people assigned to primary care of the National Health System throughout the national territory. The data collected includes, among others, health problems, and drugs prescribed and that have been dispensed in pharmacies. The data source is the electronic medical record, and the system allows anonymized use of the data. The type of sampling used is a single-stage random sampling by conglomerates (basic health areas), stratified by autonomous community and size of the municipality in which the health centers are located. Health problems are coded with ICD9, ICD10ES or CIAP2 and drugs are coded with the National Code.

## Information search and data collection

BDCAP data are organized into separate information cubes, each containing a single analysis or study variable. There are currently seven available variables (health problems, comorbidity, consultations, medications, visits, procedures, and parameters). In our study, the entry "health problems" of the statistical portal associated with the BDCAP 2019 database was used. This entry included those health problems that had been "active" in the year of study and that were registered in the clinical record in coded form.

We perform a search for the health problem "hypothyroidism/myxedema", coded as T86. To characterize the population of patients with hypothyroidism, the following demographic and socioeconomic variables were described: gender (male, female), age (large groups registered in BDCAP, that is, 0–14, 15–34, 35–64 and 65 years and over), country of birth (Spain, Europe-European Union, Eastern Mediterranean, Latin America and Others or unknown, which includes the countries of Asia, the Pacific, North America, the rest of Europe and the rest of the African countries not included in the Eastern Mediterranean), size of the municipality (≤10,000, 10,001–50,000, 50,001–100,000, 100,001–500,000 and more than 500,000 inhabitants), income level (≥100,000, 18,000–99,999, <18,000 euros/year, very low income, unclassified), and employment status (active, unemployed, pensioners, non-active, other situations).

## Subpopulations study

Different subpopulation selection filters with health problems were used. The use of filters allowed the selection of a subpopulation of users based on the health problem or problems under study. In this way we were able to quantify the population with hypothyroidism that also has another specific health problem, such as malignancy of any kind. Using the appropriate filters, we were able to characterize the population with hypothyroidism and the following specific types of malignancies: breast cancer (X76), colorectal cancer (D75), prostate cancer (Y77), hematologic cancer (leukemia, B73, and lymphoma, B72), malignant neoplasms of the respiratory system (R84 and R85), cancer of the bladder (U76), cancer of the thyroid (T71), other neoplasms of the digestive system (D77), renal cancer (U75), cancer of the cervix (X75), gastric cancer (D74) and pancreatic cancer (D76).

## Subpopulations with medication

We used the Medications cube to select the subpopulation of patients with hypothyroidism who were on replacement therapy. To do this, we employed the following filters in the drugs dimension: anatomical group H (hormones), therapeutic subgroup H03 (thyroid therapy), pharmacological subgroup (H03A, thyroid preparations), chemical subgroup (H03AA, thyroid hormones; H03BA), and active ingredients (H03AA01, sodium levothyroxine; H03AA02, sodium liothyronine; H03AA03, association of levothyroxine and sodium liothyronine; H03AA04, tiratricol; H03AA05, thyroid gland preparations). Once the subpopulation under

treatment was selected, the appropriate filters of the necessary dimensions were used to characterize its sociodemographic parameters, as well as the prevalence of malignant neoplasms.

### Statistical analysis

Quantitative variables are expressed as absolute value (people with health problems) and percentage in relation to the total population or reference subpopulation. The calculation of the proportion of malignant neoplasms in the population of patients with and without hypothyroidism was used to calculate the relative risk using the odds ratio (OR) and its 95% confidence interval (CI). The same procedure was used to calculate the relative risk of each of the malignant neoplasms studied in patients with a diagnosis of hypothyroidism, as well as to analyze the possible differences in this relative risk based on age, gender and socioeconomic characteristics of the studied subjects. Finally, the relative risk of total cancer and specific malignant neoplasms in the subpopulation of patients with hypothyroidism in replacement therapy was analyzed. The relationship between qualitative variables was studied using the chi-square test. Significant differences were considered when the P value was <0.05.

## Results

### Studied patients

In 2019, 2,414,165 people were diagnosed with hypothyroidism, which represents 5.13% of the total Spanish population (47,105,358), and 6.2% of the population with health problems registered in the BDCAP database (38,365,258 people). The percentage of hypothyroid patients was higher in women than in men and raised as the age of the subjects increased, and the level of annual income decreased. A higher frequency of hypothyroidism was also observed in residents of cities with more than 100,000 inhabitants, in those born in Latin America, and in unemployed or pensioners (Table 1).

### Overall cancer in hypothyroid patients

The total frequency of malignant neoplasms in the cohort of hypothyroid patients studied was 7.62%, which was significantly higher than that found in patients without a diagnosis of hypothyroidism (4.55%, OR 1.73 [1.72–1.74]; P<0.0001) (Table 2). The higher frequency of cancer in hypothyroid subjects was observed in both men (OR 2.15 [2.13–2.17]; P<0.0001) and women (OR 1.67 [1.636–1.68]; P<0.0001) and in subjects younger than 65 years. However, hypothyroid persons aged 65 years or older had a reduced risk of cancer (OR 0.98 [0.97–0.98]; P<0.0001). When we analyzed our population by groups of annual income level, size of the municipality, country of birth, and employment situation, the relative risk of cancer in subjects with a diagnosis of hypothyroidism was, in all cases, significantly higher than that of subjects without this diagnosis (P<0.0001).

### Specific tumor types in patients with hypothyroidism

The most common site-specific malignancies in the Spanish population in 2019 were breast in women (1.50%), prostate in men (1.23%), followed by colorectal cancer (0.67%), hematologic malignancy (0.39%), respiratory tract cancer (0.27%), and bladder cancer (0.27%) in both sexes (S1 Table).

Table 3 shows that hypothyroidism was associated with an increased risk of all the specific neoplasms studied, except for bladder cancer (OR 1.01 [0.98–1.03]; N.S.). This risk was higher for thyroid cancer (OR 5.07 [4.96–5.48]; P<0.0001), followed by cancers of the respiratory tract (OR 1.83 [1.79–1.87]; P<0.0001).

**Table 1. Total population with health problems registered in the BDCAP database in 2019 and distribution of patients with hypothyroidism.**

| | Total population in BDCAP | Patients with hypothyroidism | |
|---|---|---|---|
| | | Number | Percentage |
| **All** | 38,365,258 | 2,414,165 | 6.29 |
| **Gender** | | | |
| Male | 18,230,737 | 428,414 | 2.35 |
| Female | 20,134,521 | 1,985,751 | 9.86 |
| **Age (years)** | | | |
| 0–14 | 5,598,051 | 33,129 | 0.59 |
| 15–34 | 7,591,284 | 307,956 | 4.06 |
| 35–64 | 17,119,977 | 1,203,356 | 7.03 |
| 65 and over | 8,055,946 | 869,724 | 10.8 |
| **Income level (€/year)** | | | |
| ≥100,000 | 261,424 | 11,690 | 4.47 |
| 18,000–99,999 | 11,312,807 | 645,694 | 5.71 |
| <18,000 | 24,361,736 | 1,596,420 | 6.55 |
| Very low | 2,101,006 | 151,086 | 7.19 |
| Unclassified | 328,285 | 9,275 | 2.83 |
| **Municipality size** | | | |
| <10,000 | 5,692,080 | 333,545 | 5.86 |
| 10,001–50,000 | 11,795,541 | 716,125 | 6.07 |
| 50.001–100,000 | 5,304,250 | 313,714 | 5.91 |
| 100,001–500,000 | 9,407,737 | 624,054 | 6.63 |
| >500,000 | 6,165,651 | 426,727 | 6.92 |
| **Country of birth** | | | |
| Spain | 27,916,773 | 1,786,906 | 6.40 |
| Europe (EU) | 1,156,493 | 58,986 | 5.10 |
| Africa | 217,416 | 5,477 | 2.52 |
| Latin America | 1,765,944 | 137,389 | 7.78 |
| Eastern Mediterranean | 653,138 | 26,540 | 4.06 |
| Other or unknown | 6,655,495 | 398,867 | 5.99 |
| **Employment situation** | | | |
| Active | 14,591,010 | 842,036 | 5.77 |
| Unemployed | 2,798,342 | 208,315 | 7.44 |
| Not active | 11,268,847 | 475,044 | 4.22 |
| Pensioners | 8,502,723 | 816,313 | 9.60 |
| Other situations | 1,204,336 | 72,456 | 6.02 |

All the relative risks obtained in the different neoplasms were higher in men compared to women. In men, the highest relative risk was observed for cancers of the thyroid, respiratory tract, digestive (other), and kidney. In women, the highest risks were noticed in cancers of the thyroid, respiratory tract, digestive (other) and gastric.

Hypothyroid patients aged 65 or over did not present an increased risk of renal or digestive (other) cancer. Furthermore, these patients showed a decreased risk of cancers of the bladder (OR 0.58 [0.56–0.60]; P<0.0001), colorectal (OR 0.83 [0.81–0.84]; P<0.0001), gastric (OR 0.91 [0.87–0.96]; P = 0.0002), pancreatic (OR 0.93 [0.87–0.99]; P = 0.0327) and prostate (OR 0.97 [0.95–0.99]; P = 0.0095) (Table 3).

**Table 2. Prevalence of cancer in the entire cohort of studied subjects and in patients with and without hypothyroidism, classified by various demographic variables.**

| | All subjects | | Patients with hypothyroidism | | Patients without hypothyroidism | | | | |
|---|---|---|---|---|---|---|---|---|---|
| | No. | % | No. | % | No. | % | OR | 95% CI | P |
| **All** | 1,819,173 | 4.74 | 183,947 | 7.62 | 1,635,226 | 4.55 | 1.73 | 1.72–1.74 | <0.0001 |
| **Gender** | | | | | | | | | |
| Male | 865,161 | 4.75 | 40,496 | 9.45 | 824,665 | 4.63 | 2.15 | 2.13–2.17 | <0.0001 |
| Female | 954,013 | 4.74 | 143,451 | 7.22 | 810,562 | 4.47 | 1.67 | 1.66–1.68 | <0.0001 |
| **Age (years)** | | | | | | | | | |
| 0–14 | 14,259 | 0.25 | 341 | 1.03 | 13,918 | 0.25 | 4.15 | 3.72–4.62 | <0.0001 |
| 15–34 | 51,558 | 0.68 | 3,482 | 1.13 | 48,076 | 0.66 | 1.72 | 1.66–1.78 | <0.0001 |
| 35–64 | 649,489 | 3.79 | 63,219 | 5.25 | 586,270 | 3.68 | 1.45 | 1.44–1.46 | <0.0001 |
| 65 and over | 1,103,868 | 13.7 | 116,905 | 13.44 | 986,963 | 13.73 | 0.98 | 0.97–0.98 | <0.0001 |
| **Income level (€/year)** | | | | | | | | | |
| ≥100,000 | 13,438 | 5.14 | 1,142 | 9.77 | 12,296 | 4.92 | 2.09 | 1.96–2.23 | <0.0001 |
| 18,000–99,999 | 575,869 | 5.09 | 52,627 | 8.15 | 523,242 | 4.91 | 1.72 | 1.70–1.74 | <0.0001 |
| <18,000 | 1,144,506 | 4.7 | 119,768 | 7.5 | 1,024,738 | 4.5 | 1.72 | 1.71–1.73 | <0.0001 |
| Very low | 75,365 | 3.59 | 9,562 | 6.33 | 65,803 | 3.37 | 1.93 | 1.89–1.98 | <0.0001 |
| Unclassified | 9,994 | 3.04 | 848 | 9.14 | 9,146 | 2.87 | 3.41 | 3.17–3.67 | <0.0001 |
| **Municipality size** | | | | | | | | | |
| <10,000 | 269,707 | 4.74 | 25,087 | 7.52 | 244,620 | 4.57 | 1.70 | 1.68–1.72 | <0.0001 |
| 10,001–50,000 | 499,563 | 4.24 | 50,169 | 7.01 | 449,394 | 4.06 | 1.78 | 1.77–1.80 | <0.0001 |
| 50.001–100,000 | 226,616 | 4.27 | 21,481 | 6.85 | 205,135 | 4.11 | 1.71 | 1.69–1.74 | <0.0001 |
| 100,001–500,000 | 476,839 | 5.07 | 49,292 | 7.9 | 427,547 | 4.87 | 1.68 | 1.66–1.69 | <0.0001 |
| >500,000 | 346,447 | 5.62 | 37,919 | 8.89 | 308,528 | 5.38 | 1.72 | 1.70–1.74 | <0.0001 |
| **Country of birth** | | | | | | | | | |
| Spain | 1,386,429 | 4.97 | 143,509 | 8.03 | 1,242,920 | 4.76 | 1.75 | 1.74–1.76 | <0.0001 |
| Europe (EU) | 37,068 | 3.21 | 3,576 | 6.06 | 33,492 | 3.05 | 2.05 | 1.98–2.55 | <0.0001 |
| Africa | 2,580 | 1.19 | | | 2,580 | 1.22 | NA | | |
| Latin America | 35,633 | 2.02 | 4,698 | 3.42 | 30,935 | 1.9 | 1.83 | 1.77–1.89 | <0.0001 |
| Eastern Mediterranean | 8,309 | 1.27 | 745 | 2.81 | 7,564 | 1.21 | 2.36 | 2.19–2.55 | <0.0001 |
| Other or unknown | 349,154 | 5.25 | 31,288 | 7.84 | 317,866 | 5.08 | 1.59 | 1.57–1.61 | <0.0001 |
| **Employment situation** | | | | | | | | | |
| Active | 372,317 | 2.55 | 31,893 | 3.79 | 340,424 | 2.48 | 1.55 | 1.53–1.57 | <0.0001 |
| Unemployed | 78,216 | 2.8 | 8,148 | 3.91 | 70,068 | 2.71 | 1.46 | 1.43–1.50 | <0.0001 |
| Not active | 216,083 | 1.92 | 30,111 | 6.34 | 185,972 | 1.72 | 3.86 | 3.81–3.91 | <0.0001 |
| Pensioners | 1,100,910 | 12.95 | 108,573 | 13.3 | 992,337 | 12.91 | 1.03 | 1.03–1.04 | <0.0001 |
| Other situations | 51,647 | 4.29 | 5,222 | 7.21 | 46,425 | 4.1 | 1.82 | 1.76–1.87 | <0.0001 |

Abbreviations: NA, not available.

## Influence of socioeconomic variables

Income level had limited influence on the association of hypothyroidism and cancer, since the different groups studied showed an increased risk of all tumors, except for bladder cancer (Fig 1).

Residents in municipalities with 100,001 to 500,000 inhabitants had a reduced risk of bladder cancer (OR 0.92 [0.88–0.97]; P = 0.0017); however, residents in municipalities with 50,001 to 100,000 inhabitants had an increased risk of this tumor (OR 1.16 [1.08–1.24]; P<0.0001).

**Table 3. Relative risk of different types of malignancies in patients with hypothyroidism classified by age and sex.**

|  | | Gender | | Age (years) | | |
|---|---|---|---|---|---|---|
|  | All | Male | Female | 15–34 | 35–64 | 65 and over |
| **Breast** | 1.49 (1.48–1.51) P<0.0001 | --- | 1.49 (1.48–1.51) P<0.0001 | --- | 1.03 (1.02–1.05) P = 0.0001 | 1.03 (1.02–1.05) P<0.0001 |
| **Colorectal** | 1.42 (1.40–1.44) P<0.0001 | 1.62 (1.58–1.67) P<0.0001 | 1.58 (1.55–1.60) P<0.0001 | --- | 1.17 (1.09–1.14) P<0.0001 | 0.83 (0.81–0.84) P<0.0001 |
| **Prostate** | 1.76 (1.72–1.80) P<0.0001 | 1.76 (1.72–1.80) P<0.0001 | --- | --- | 1.16 (1.08–1.24) P<0.0001 | 0.97 (0.95–0.99) P = 0.0095 |
| **Hematologic** | 1.74 (1.71–1.77) P<0.0001 | 2.25 (2.18–2.33) P<0.0001 | 1.74 (1.70–1.77) P<0.0001 | 1.68 (1.59–1.78) P<0.0001 | 1.49 (1.45–1.53) P<0.0001 | 1.21 (1.18–1.23) P<0.0001 |
| **Respiratory tract** | 1.83 (1.79–1.87) P<0.0001 | 3.74 (3.65–3.84) P<0.0001 | 1.94 (1.88–1.99) P<0.0001 | --- | 1.50 (1.45–1.55) P<0.0001 | 1.08 (1.05–1.10) P<0.0001 |
| **Bladder** | 1.01 (0.98–1.03) 1.02 P = 0.5135 | 1.78 (1.72–1.84) P<0.0001 | 1.73 (1.67–1.80) P<0.0001 | --- | 0.79 (0.75–0.83) P<0.0001 | 0.58 (0.56–0.60) P<0.0001 |
| **Thyroid** | 5.07 (4.96–5.18) P<0.0001 | 9.58 (9.09–10.10) P<0.0001 | 3.41 (3.33–3.49) P<0.0001 | 6.28 (5.72–6.90) P<0.0001 | 4.24 (4.12–4.36) P<0.0001 | 3.27 (3.16–3.40) P<0.0001 |
| **Digestive (other)** | 1.58 (1.53–1.63) P<0.0001 | 2.35 (2.23–2.48) P<0.0001 | 1.84 (1.77–1.92) P<0.0001 | --- | 1.22 (1.16–1.29) P<0.0001 | 0.97 (0.93–1.01) P = 0.1257 |
| **Renal** | 1.52 (1.47–1.57) P<0.0001 | 2.35 (2.22–2.48) P<0.0001 | 1.71 (1.64–1.79) P<0.0001 | --- | 1.05 (0.98–1.12) P = 0.1536 | 0.98 (0.94–1.02) P = 0.2750 |
| **Cervix** | 1.29 (1.25–1.33) P<0.0001 | --- | 1.29 (1.25–1.33) P<0.0001 | 1.28 (1.12–1.45) P = 0.0001 | 1.03 (0.99–1.07) P = 0.0945 | 1.07 (1.01–1.13) P = 0.0304 |
| **Gastric** | 1.54 (1.48–1.61) P<0.0001 | 1.78 (1.63–1.93) P<0.0001 | 1.77 (1.68–1.86) P<0.0001 | --- | 1.20 (1.10–1.30) P<0.0001 | 0.91 (0.87–0.96) P = 0.0002 |
| **Pancreas** | 1.59 (1.50–1.68) P<0.0001 | 1.98 (1.78–2.22) P<0.0001 | 1.59 (1.49–1.70) P<0.0001 | --- | 1.24 (1.12–1.67) P<0.0001 | 0.93 (0.87–0.99) P = 0.0327 |

Data are OR with 95% CI, and P value.

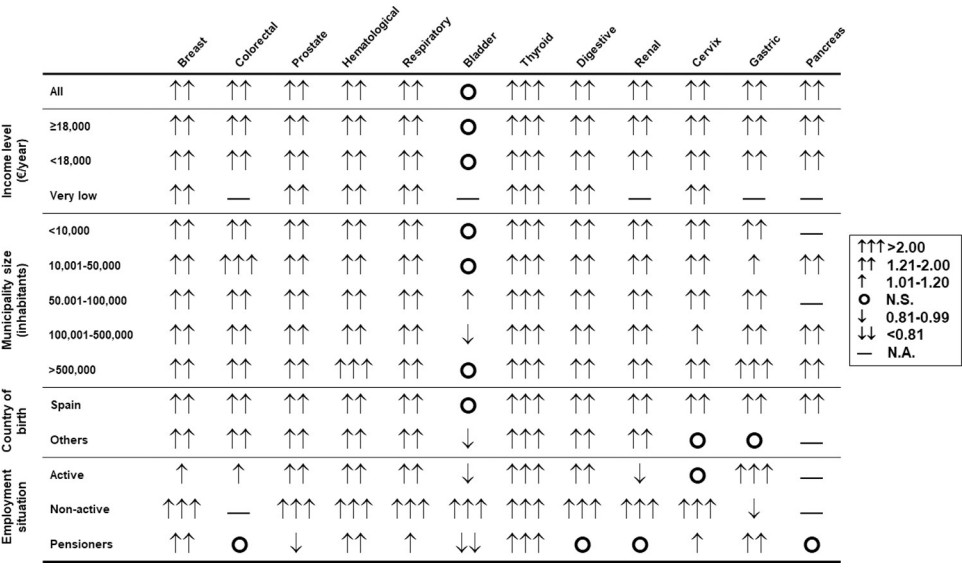

**Fig 1. Relative risk (OR) of the different malignancies studied in patients with hypothyroidism classified according to sociodemographic variables.** Arrows are used to indicate a statistically significant OR value greater than (↑) or less than 1 (↓). The circle (○) indicates a non-significant OR value (N.S.). The dash (—) is used to represent data not available (N.A.).

The risk associated with the rest of the studied tumors was not affected by the size of the municipality.

Hypothyroid patients born in Spain had an increased risk of all cancers except bladder cancer. However, those born in other countries had a reduced risk of bladder cancer and a neutral risk of cervical and gastric cancer.

Lastly, the employment status had an influence on the risk of cancer, since working people had a decreased risk of bladder and renal cancer, while non-active people had an increase in all tumors with available data (including bladder cancer) and a decreased risk of gastric cancer. Pensioners showed a decreased risk for bladder and prostate cancers, a neutral risk for colorectal, kidney, and pancreatic cancers, and an elevated risk for the remaining malignancies (S2 Table).

## Cancer risk in patients treated with thyroid hormone

Of the 2,282,124 patients diagnosed with hypothyroidism in 2018, BDCAP shows that there were 977,761 (42.84%) subjects with thyroid hormone replacement therapy (Table 4). The frequency of malignancies in treated patients was 8.99%, while in untreated patients it was 6.36%, with implied an OR of 1.30 (1.28–1.31) (P<0.0001).

The relative risk of cancer in patients receiving hormone replacement therapy compared with patients without treatment was higher in men (OR 1.53 [1.49–1.56]; P<0.0001) than in women (OR 1.28 [1.26–1.29]; P<0.0001). This increased risk was also seen in patients 65 years of age and older (OR 1.03 [1.02–1.04]; P<0.0001), but was quantitatively more moderate than the same risk in younger patients (Table 4). In addition, when the patients were classified into different groups of socioeconomic variables, it was observed that patients treated with thyroid hormone always exhibited a higher risk of cancer than the untreated patients (Table 4).

## Trend of cancer frequency in hypothyroid patients

Fig 2 shows the number of people with cancer per 1,000 attended in the Spanish health system and registered in BDCAP from 2011 to 2019. The trend has been upward in the general population, without major differences between men and women. In people with hypothyroidism, the upward trend has been more marked than in the general population in both sexes. In addition, hypothyroid men presented higher values than women in all years.

## Discussion

To our knowledge, this is the first study that analyzes the risk of cancer in Spanish hypothyroid people using the BDCAP database. The main findings are: (a) the relative risk of total cancer and site-specific cancer (with the exception of bladder cancer) is increased in Spanish people with hypothyroidism; (b) this increased risk is not affected by gender or the studied socioeconomic characteristics; (c) in subjects 65 years of age or older, there is a decreased risk of overall cancer, as well as colorectal, prostate, bladder, gastric, and pancreatic cancers; (d) socioeconomic variables have limited influence on overall cancer risk and risk of specific cancers; (e) patients receiving thyroid hormone replacement therapy have a higher risk of overall cancer compared to untreated hypothyroid patients.

The three more common tumors in the Spanish population (breast, colorectal and prostate) have a higher risk in hypothyroid people, but only the breast cancer maintains the risk in all age groups. These results are in essential agreement with our previous study, carried out in a hospital cohort of approximately half a million subjects, using big data methodology, in which we showed that there was a significant association between the diagnosis of hypothyroidism and cancer, although this association was less evident in subjects older than 60 years [20].

**Table 4. Prevalence of cancer in patients with hypothyroidism classified according to the presence or absence of treatment with thyroid hormones (data of 2018).**

| | Patients with hypothyroidism | | Prevalence of cancer in hypothyroid patients | | | OR (95% CI) | P |
|---|---|---|---|---|---|---|---|
| | All patients | Patients with treatment | All patients | Patients with treatment | Patients without treatment | | |
| **All** | 2,282,124 | 977,761 (42.84) | 162,068 (7.10) | 79,127 (8.09) | 82,941 (6.36) | 1.30 (1.28–1.31) | <0.0001 |
| **Gender** | | | | | | | |
| Male | 397,073 | 137,440 (34.61) | 35,275 (8.88) | 15,418 (11.22) | 19,857 (7.65) | 1.53 (1.49–1.56) | <0.0001 |
| Female | 1,885,051 | 840,320 (44.58) | 126,793 (6.73) | 63,709 (7.58) | 63,084 (6.04) | 1.28 (1.26–1.29) | <0.0001 |
| **Age (years)** | | | | | | | |
| 0–14 | 31,460 | 5,459 (17.35) | 0 (0) | 0 (0) | 0 (0) | | <0.0001 |
| 15–34 | 295,107 | 81,658 (27.67) | 3,398 (1.15) | 1,210 (1.48) | 2,188 (1.03) | 1.45 (1.35–1.56) | <0.0001 |
| 35–64 | 1,145,421 | 503,753 (43.98) | 57,420 (5.01) | 28,995 (5.76) | 28,425 (4.43) | 1.32 (1.30–1.34) | <0.0001 |
| 65 and over | 810,136 | 386,891 (47.76) | 100,995 (12.47) | 48,856 (12.63) | 52,139 (12.32) | 1.03 (1.02–1.04) | <0.0001 |
| **Income level (€/year)** | | | | | | | |
| ≥100,000 | 8,926 | 3,842 (43.04) | 751 (8.41) | 473 (12.31) | 278 (5.47) | 2.43 (2.08–2.83) | <0.0001 |
| 18,000–99,999 | 519,947 | 222,412 (42.78) | 39,739 (7.64) | 19,365 (8.71) | 20,374 (6.85) | 1.30 (1.27–1.32) | <0.0001 |
| <18,000 | 1,600,629 | 682,254 (42.62) | 112,454 (7.03) | 54,576 (8.00) | 57,878 (6.30) | 1.29 (1.28–1.31) | <0.0001 |
| Very low | 144,909 | 66,942 (46.20) | 8,563 (5.91) | 4,456 (6.66) | 4,107 (5.27) | 1.28 (1.23–1.34) | <0.0001 |
| Unclassified | 7,713 | 2,310 (29.95) | 562 (7.29) | 256 (11.08) | 306 (5.66) | 2.08 (1.74–2.47) | <0.0001 |
| **Municipality size** | | | | | | | |
| <10,000 | 314,191 | 145,548 (46.32) | 21,703 (6.91) | 10,602 (7.28) | 11,101 (6.58) | 1.11 (1.08–1.15) | <0.0001 |
| 10,001–50,000 | 680,563 | 257,748 (37.87) | 44,053 (6.47) | 17,894 (6.94) | 26,159 (6.19) | 1.13 (1.11–1.15) | <0.0001 |
| 50.001–100,000 | 297,184 | 138,059 (46.46) | 18,805 (6.33) | 9,802 (7.10) | 9,003 (5.66) | 1.27 (1.24–1.31) | <0.0001 |
| 100,001–500,000 | 589,297 | 268,109 (45.50) | 43,468 (7.38) | 23,940 (8.93) | 19,528 (6.08) | 1.51 (1.49–1.54) | <0.0001 |
| >500,000 | 400,889 | 168,297 (41.98) | 34,040 (8.49) | 16,890 (10.04) | 17,150 (7.37) | 1.40 (1.37–1.43) | <0.0001 |
| **Country of birth** | | | | | | | |
| Spain | 1,689,050 | 757,544 (44.85) | 127,038 (7.52) | 64,874 (8.56) | 62,164 (6.67) | 1.31 (1.29–1.32) | <0.0001 |
| Europe (EU) | 53,190 | 19,474 (36.61) | 2,854 (5.37) | 1,136 (5.83) | 1,718 (5.10) | 1.15 (1.07–1.25) | 0.0003 |
| Africa | 4,672 | 1,319 (28.23) | 0 (0) | 0 (0) | 0 (0) | | |
| Latin America | 121,188 | 46,966 (38.75) | 3,988 (3.29) | 1,990 (4.24) | 1,998 (2.69) | 1.60 (1.50–1.70) | <0.0001 |
| Eastern Mediterranean | 22,767 | 8,504 (37.35) | 611 (2.68) | 317 (3.73) | 294 (2.06) | 1.84 (1.57–2.16) | <0.0001 |
| Other or unknown | 391,256 | 143,953 (36.79) | 27,471 (7.02) | 10,767 (7.48) | 16,704 (6.75) | 1.12 (1.09–1.14) | <0.0001 |
| **Employment situation** | | | | | | | |

(*Continued*)

**Table 4.** (Continued)

| | Patients with hypothyroidism | | Prevalence of cancer in hypothyroid patients | | | OR (95% CI) | P |
|---|---|---|---|---|---|---|---|
| | All patients | Patients with treatment | All patients | Patients with treatment | Patients without treatment | | |
| Active | 784,701 | 322,008 (41.04) | 28,194 (3.59) | 14,475 (4.96) | 13,719 (2.97) | 1.54 (1.50–1.58) | <0.0001 |
| Unemployed | 197,327 | 84,835 (42.99) | 7,160 (3.62) | 3,708 (4.37) | 3,452 (3.07) | 1.44 (1.38–1.51) | <0.0001 |
| Not active | 474,823 | 194,993 (41.07) | 28,069 (5.91) | 14,480 (7.43) | 13,589 (4.86) | 1.57 (1.53–1.61) | <0.0001 |
| Pensioners | 766,274 | 359,743 (46.95) | 94,406 (12.32) | 45,205 (12.57) | 49,201 (12.10) | 1.04 (1.03–1.06) | <0.0001 |
| Other situations | 58,999 | 16,182 (27.43) | 4,239 (7.18) | 1,259 (7.78) | 2,980 (6.96) | 1.13 (1.05–1.21) | 0.0006 |

Data are the absolute values of patients and the percentage with respect to the total number of people in their group or subgroup.

Abbreviations: NA, not available.

Furthermore, a recent systematic review of controlled clinical trials and observational studies found no association between subclinical hypothyroidism and breast and prostate cancers [21]. Another systematic review and meta-analysis of 15 observational studies [22] showed that, compared to euthyroid subjects, hypothyroidism was associated with an increased risk of thyroid cancer in the first 10 years after diagnosis of deficiency thyroid hormone.

Initial investigations found that patients with hypothyroidism had higher risk of breast cancer [23, 24]. However, other studies showed that hypothyroidism was associated with a reduced risk of this tumor [21, 25] or that there was no significant relationship [13, 17, 26–31].

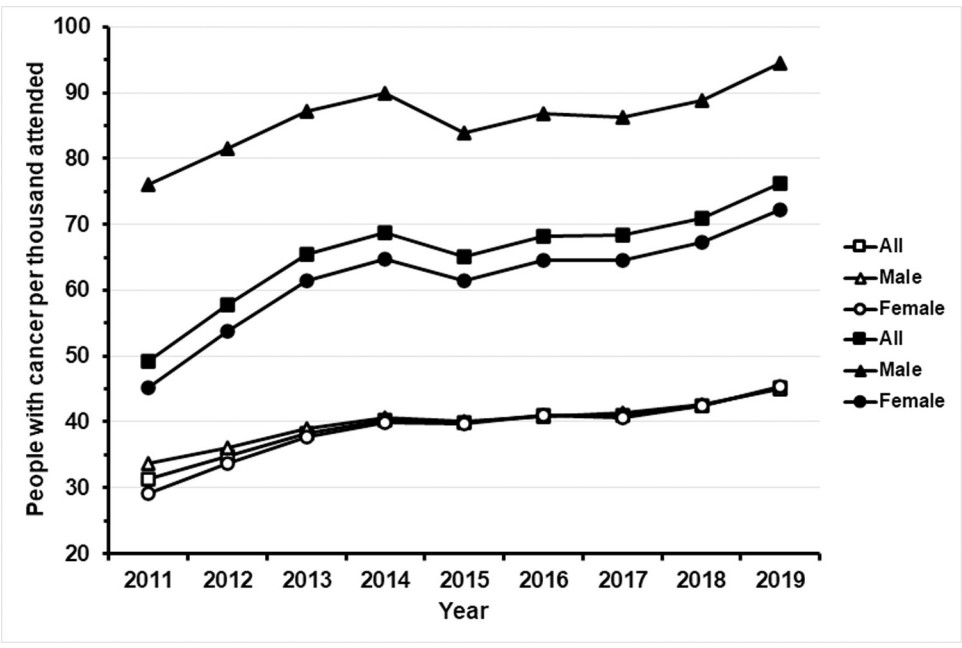

**Fig 2. Frequency of cancer in the total population registered in the BDCAP database and in patients with hypothyroidism from 2011 to 2019.** Data are expressed in number of people per 1,000 attended. The triangles represent the data for men, the circles those for women and squares those of patients of both sexes. The empty symbols refer to the total population, while the filled symbols represent patients with hypothyroidism.

A meta-analysis including 12 studies showed that there was a 6% increase in risk for breast cancer among women with primary hypothyroidism [17]. Another analysis showed a significant inverse association between invasive breast cancer and history of hypothyroidism [32]. Consistent with our results, two national database studies, in Taiwan [33] and the Netherlands [14], showed that women with hypothyroidism had a higher risk of breast cancer than the non-hypothyroid.

Boursi et al. [16] showed that untreated hypothyroidism was associated with elevated risk of colorectal cancer. Mu et al. [34] found that the prevalence of hypothyroidism was significantly higher in subjects with colorectal cancer compared to controls. In line with these investigations our results showed an increased risk of colorectal cancer in hypothyroid people younger than 65 years. Hypothyroidism also behaved as a risk factor for prostate cancer in our population and as a protective factor in men over 65 years of age and pensioners. These findings are in line to those reported by Mondul et al. [35], who showed that men with hypothyroidism had a lower risk of prostate cancer relative to euthyroid men, although the mean age in this study was 57 years. The relationship between thyroid hypofunction and thyroid cancer has been demonstrated in different studies [21, 22, 36], as well as in our previous report [20]. Accordingly, the present study shows the highest relative risks in thyroid cancer and that the relationship between hypothyroidism and thyroid cancer is maintained in all age groups and in patients with different socioeconomic conditions.

Previous literature offers limited information on other cancers [12, 21, 37, 38]. In our population, hematological and respiratory tract cancers exhibited a homogeneous behavior, that is, their risk was not significantly modified by gender, age, or socioeconomic characteristics. Subgroup analysis showed that bladder cancer had a peculiar behavior that was difficult to explain. The relative risk was increased in non-active persons and inhabitants of municipalities with 50,001 to 100,000 inhabitants, but reduced in active persons, pensioners and inhabitants of municipalities with 100,001 to 500,000 inhabitants. Our result contrasts with that reported by Mellemgaard et al [37] in a cohort of individuals who were discharged from a Danish hospital. These authors found an increased risk for bladder cancer among women. Differences may be accounted for by the fact that admitted patients may differ from the general population registered in BDCAP. Recently it has been shown that thyroid receptor-interacting protein 13 (TRIP13), a protein associated with the progression of several cancers, promotes proliferation and invasion of bladder cancer [39]. It can be speculated that in certain population groups, thyroid hormone deficiency might act by modifying the effect of certain proteins or carcinogen factors on the genesis of bladder neoplasms.

Previous clinical findings have supported that hypothyroidism predisposes to hepatocellular carcinoma development [40, 41]. Our data suggest a risk for hepatocellular cancer (included in other digestive cancers) that is not affected by gender or socioeconomic conditions, but is affected by age, since the risk disappeared in people older than 65 years. Cervical cancer risk was not significantly affected by age and was only reversed in non-working women. Interestingly, our data suggest a possible protective effect from hypothyroidism against colorectal, prostate, bladder, gastric and pancreatic cancer in people over age 65 elderly. The reduced risk of these malignancies in the elderly could be explained by the fact that thyroid hormone deficiency might slow down tumor development and progression, as has been suggested by some authors [42].

Although thyroid hormone replacement has been reported to have a protective effect against some cancers [16, 33], in our study the risk of all cancers was higher in patients treated with thyroid hormone. This surprising finding could be accounted for by the fact that the subgroup of patients in replacement treatment has a more marked degree of thyroid hormone deficiency and, therefore, a greater predisposition to cancer. It should be noted that less than

half of the hypothyroid patients were on replacement therapy. The BDCAP database does not allow knowing the criteria for treatment with thyroid hormone, so it is conceivable that a large part of untreated hypothyroid subjects presents only mild or transient subclinical hypothyroidism without the need for treatment. In addition, binding of thyroid hormones to their receptors could activate different pathways, such as β-catenin, PI3K and MAPK/ERK1/2, leading to increased tumor cell proliferation and angiogenesis [43]. There is also evidence of crosstalk between thyroid hormones and the estrogen signaling pathway via estrogen receptors, suggesting a possible role of thyroid hormones in breast cancer [44].

The main strength is the large sample size and an adequate distribution of the population studied, since most of the Spanish population receives healthcare through general practitioners of the public health system. The BDCAP database is a nationwide database containing data of most of the Spanish population and contains all active health problems, thus allowing an accurate assessment of disease prevalences and drug prescriptions in each of the years it is available, without data selection bias. Although our study has detected some statistically significant links between hypothyroidism and various malignancies, BDCAP lacks information regarding tumor staging, duration of tumor disease or hypothyroidism, as well as the degree of control of thyroid hypofunction in patients receiving replacement therapy. Some relevant confounding factors in carcinogenesis, such as family history, dietary or pharmacological factors, could not be considered. Our study was unable to provide data on antithyroid antibodies, iodine intake, or the duration of replacement therapy in treated patients. We were also unable to know the etiology, duration or degree of hypothyroidism in our patients, since this information is not found in the BDCAP.

We think that our results are relevant for health care planning not only because of the demonstration of an increased risk of cancer in hypothyroid people, but also because the trend of this relationship in the last 10 years has been upward, especially among men, as Fig 2 shows. Data herein reported allow us to suggest that screening programs currently in force in the general population should be carried out with greater emphasis on hypothyroid people.

## Conclusions

In summary, we have shown for the first time in the Spanish population a robust and significant association between hypothyroidism and total cancer. The increased risk in hypothyroid individuals is especially marked for thyroid, respiratory tract, prostate, and hematologic cancers. However, people over 65 years of age with hypothyroidism have a neutral risk of some cancers and a reduced risk of others, including common tumors such as prostate and colorectal. Taken together, our results suggest that patients with hypothyroidism should be routinely followed up for common cancers and that prospective studies should be performed in hypothyroid persons to clarify a potential association between hypothyroidism and site-specific malignancies.

## Supporting information

**S1 Table. Prevalence of the different malignant neoplasms studied in the Spanish population in 2019.**
(DOCX)

**S2 Table. Relative risk of the different site-specific malignancies studied in patients with hypothyroidism classified according to socioeconomic variables.**
(DOCX)

**S1 File. Prevalence of cancer in the total Spanish population and in patients with hypothyroidism.**
(XLSX)

**S2 File. Prevalence of cancer in Spanish patients with hypothyroidism on replacement therapy.**
(XLSX)

**S3 File. Trend of cancer frequency in Spanish hypothyroid patients from 2011 to 2019.**
(XLSX)

## Author Contributions

**Conceptualization:** Juan J. Díez.

**Data curation:** Juan J. Díez.

**Formal analysis:** Juan J. Díez.

**Investigation:** Juan J. Díez.

**Methodology:** Juan J. Díez.

**Project administration:** Juan J. Díez.

**Software:** Juan J. Díez.

**Supervision:** Juan J. Díez, Pedro Iglesias.

**Validation:** Juan J. Díez, Pedro Iglesias.

**Visualization:** Juan J. Díez, Pedro Iglesias.

**Writing – original draft:** Juan J. Díez.

**Writing – review & editing:** Juan J. Díez, Pedro Iglesias.

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
