## [Decision Letter · Decision Letter 0]

11 Sep 2022

PONE-D-22-16535Malignant neoplasms in people with hypothyroidism in Spain: a population-based analysisPLOS ONE

Dear Dr. Díez, 

Thank you for submitting your manuscript to PLOS ONE. After careful consideration, we feel that it has merit but does not fully meet PLOS ONE’s publication criteria as it currently stands. Therefore, we invite you to submit a revised version of the manuscript that addresses the points raised during the review process.

We look forward to receiving your revised manuscript.

Kind regards,

Donovan Anthony McGrowder, PhD., MA., MSc

Academic Editor

PLOS ONE

Journal Requirements:

Additional Editor Comments:

Dear Dr. Diez,  

Your manuscript “Malignant neoplasms in people with hypothyroidism in Spain: a population-based analysis” has been assessed by our reviewers. They have raised a number of points which we believe would improve the manuscript and may allow a revised version to be published in PLOS ONE. Their reports, together with any other comments, are below.

If you are able to fully address these points, we would encourage you to submit a revised manuscript to PLOS ONE.

Regards,

Dr. Donovan McGrowder

Associate Editor

Reviewers' comments:

Reviewer's Responses to Questions

**Comments to the Author**

1. Is the manuscript technically sound, and do the data support the conclusions?

Reviewer #1: Yes

Reviewer #2: Yes

2. Has the statistical analysis been performed appropriately and rigorously? 

Reviewer #1: Yes

Reviewer #2: Yes

3. Have the authors made all data underlying the findings in their manuscript fully available?

Reviewer #1: Yes

Reviewer #2: Yes

4. Is the manuscript presented in an intelligible fashion and written in standard English?

Reviewer #1: Yes

Reviewer #2: Yes

5. Review Comments to the Author

Reviewer #1: This large database is very well-written and give some interesting data about the relationship of various cancers with hypothyroidism. Only few minor suggestions had been advised.

1.Abstract - 'scarce influence' in the part of results should be changed into the another wordings

2.Intro - citation of estimated at 8.8% should be cited with standard format of reference

3.Intro - first paragraph 'thyroid hypofunction' should be replaced with 'hypothyroidism'

4.Page 9 - 'scarce influence' in the part of results should be changed into the another wordings

5.Page 10 - 3rd paragraph cite the reason why only less than 50% of patients diagnosed with hypothyroidism had been treated with LT4

6.Discussions were too long and should be more concise especially page 13-15

7.Ref 20 refers to subclinical hypothyroidism only

8.Ref 26 did not congruent with the written text

9.Postulated mechanism why CA bladder did not associate with hypothyroidism should be discussed.

Reviewer #2: The study, based on a large sample, found that hypothyroidism increases the risk of a variety of malignancies. However, there are still two things that can be considered. First, what are the age groups based on. Second, whether there was a difference in the degree of hypothyroidism between the alternative treatment group and the non-alternative treatment group.

---

## [Author Response · Author response to Decision Letter 0]

15 Sep 2022

COMMENTS TO THE EDITOR AND REVIEWERS

Manuscript ID: PONE-D-22-16535

Title: "Malignant neoplasms in people with hypothyroidism in Spain: a population-based analysis (REVISED VERSION R1)"

Authors: J.J. Díez et al.

GENERAL COMMENTS

Thank you very much for your email dated on September 12th, 2022, that let us know that our manuscript has merit but does not fully meet PLOS ONE’s publication criteria as it currently stands.

Following the recommendations of editor and reviewers, we have made a thorough revision of the article and we have introduced a series of modifications according to the suggestions.

All the authors would like to thank the editors and reviewers for their effort and suggestions, which has improved the quality of our manuscript.

In the new version of the article, the additions and changes have been highlighted in yellow.

RESPONSES TO THE REVIEWER’S COMMENTS

JOURNAL REQUIREMENTS

REQUIREMENT 1 

RESPONSE

Thank you for this comment. We have followed the PLOS ONE’s style requirements. Accordingly, we have renamed the files according to the instructions.

We have used Level 1 (18pt font) heading for all major sections (Abstract, Introduction, Materials and methods, Results, Discussion). We have used Level 2 (16pt font) headings for sub-sections of major sections.

Figures are sent in tif format in the revised version of the manuscript.

Each figure caption has been placed directly after the paragraph in which they are first cited.

Tables have been included directly after the paragraph in which they are first cited.

We have used the format “S1 Table” and “S2 Table” for supporting information citations.

We have cited the references in brackets.

REQUIREMENT 2

RESPONSE

Thank you for this comment. 

Our study is based on a large public database of the Ministry of Health of the Government of Spain. The study does not use personal data of any subject. The information in the database uses anonymized data and it is not possible to have individual data of any study subject.

In this study it is not possible to obtain informed consent from any patient. Nonetheless, our study was evaluated by the ethics committee of the Hospital Universitario Puerta de Hierro Majadahonda and obtained a positive evaluation. The ethics committee considered our methodology to obtain the information adequate and accepted the consent waiver proposed for this study.

In the Material and methods section of our manuscript we have included a brief statement on the approval of the ethics committee

This study was approved by the local ethics committee of the Hospital Universitario Puerta de Hierro Majadahonda (Madrid, Spain) (PI 94/22). Since our study was carried out through a database with accumulated information, the need for consent was waived by the ethics committee

REQUIREMENT 3

Upon re-submitting your revised manuscript, please upload your study’s minimal underlying data set as either Supporting Information files or to a stable, public repository and include the relevant URLs, DOIs, or accession numbers within your revised cover letter. For a list of acceptable repositories, please see http://journals.plos.org/plosone/s/data-availability#loc-recommended-repositories.Any potentially identifying patient information must be fully anonymized.

RESPONSE

Thanks for this observation. In the revised version of the manuscript we have included all the data underlying the calculations and results of our study. The information is included in three Excel files listed in the Supporting information.

The first file (S1 file) contains cancer prevalence data in the total Spanish population and in patients with hypothyroidism, as well as the different prevalences of the malignant neoplasms studied in this article. The second file (S2 file) contains the prevalence of cancer in Spanish patients with hypothyroidism who followed thyroid hormone replacement therapy. Lastly, the third file (S3 file) collects data on the frequency of cancers in the Spanish population between 2011 and 2019, as support for Figure 2 of our article.

We have included the following sentences in the revised version of the manuscript:

S1 File. Prevalence of cancer in the total Spanish population and in patients with hypothyroidism (XLSX).

S2 File. Prevalence of cancer in Spanish patients with hypothyroidism on replacement therapy (XLSX).

S3 File. Trend of cancer frequency in Spanish hypothyroid patients from 2011 to 2019 (XLSX).

Furthermore, in our Data Availability statement, we have specified the following:

All data underlying the results described in this manuscript are fully available and can be found in the Supporting information files.

REQUIREMENT 4

RESPONSE

Thanks for this comment. Certainly, as discussed in Requirement 3, we need to modify our Data Availability Statement. This is so because all the information necessary to obtain the results and all the data used for the calculations have been included in the Supporting Information in the form of three Excel files. Please amend and update our Data Availability Statement.

REQUIREMENT 5

RESPONSE

According to this requirement, we have deleted our ethics statement except for the Methods section.

REQUIREMENT 6

RESPONSE

We have had to make some modifications to our reference list based on reviewer feedback.

We have had to add a new citation based on comment 5.2 from reviewer #1. The reference ‘Santos Palacios et al., 2018’ is now the new ref. #3.

Santos Palacios S, Llavero Valero M, Brugos-Larumbe A, Díez JJ, Guillén-Grima F, Galofré JC. Prevalence of thyroid dysfunction in a Large Southern European Population. Analysis of modulatory factors. The APNA study. Clin Endocrinol (Oxf). 2018 Sep;89(3):367-375. doi: 10.1111/cen.13764.

We have omitted the reference #26 (Ditsch et al., 2010) according to the comment 5.8 from reviewer #1.

26. Ditsch N, Liebhardt S, Von Koch F, Lenhard M, Vogeser M, Spitzweg C, Gallwas J & Toth B. Thyroid function in breast cancer patients. Anticancer Research 2010 30 1713–1717.

We have omitted reference #24 (Vorherr, 1978) and #45 (Boelaert et al., 2006) as a consequence of the shortening of the discussion suggested by reviewer #1 in comment 5.6.

Vorherr H. Thyroid disease in relation to breast cancer. Klinische Wochenschrift 1978 56 1139–1145. (doi:10.1007/BF01476857)

Boelaert K, Horacek J, Holder RL, Watkinson JC, Sheppard MC, Franklyn JA. Serum thyrotropin concentration as a novel predictor of malignancy in thyroid nodules investigated by fine-needle aspiration. J Clin Endocrinol Metab. 2006 Nov;91(11):4295-301. doi: 10.1210/jc.2006-0527.

We have added a new bibliographic reference as a result of our response to comment 5.9 from reviewer #1.

Niu L, Gao Z, Cui Y, Yang X, Li H. Thyroid Receptor-Interacting Protein 13 is Correlated with Progression and Poor Prognosis in Bladder Cancer. Med Sci Monit. 2019 Sep 5;25:6660-6668. doi: 10.12659/MSM.917112.

The numbering of the rest of the references has been modified in accordance with this addition.

Lastly, we have reviewed the reference list and believe it to be complete and correct. 

ADITIONAL EDITOR COMMENTS

REVIEWERS’ COMMENTS

COMMENT 1

1.Is the manuscript technically sound, and do the data support the conclusions?

The manuscript must describe a technically sound piece of scientific research with data that supports the conclusions. Experiments must have been conducted rigorously, with appropriate controls,replication, and sample sizes. The conclusions must be drawn appropriately based on the data presented.

Reviewer #1: Yes

Reviewer #2: Yes

RESPONSE

Thank you for this comment.

COMMENT 2

2. Has the statistical analysis been performed appropriately and rigorously?

Reviewer #1: Yes

Reviewer #2: Yes

RESPONSE

Thank you.

COMMENT 3

3. Have the authors made all data underlying the findings in their manuscript fully available?

The PLOS Data policy requires authors to make all data underlying the findings described in their manuscript fully available without restriction, with rare exception (please refer to the DataAvailability Statement in the manuscript PDF file). The data should be provided as part of the manuscript or its supporting information, or deposited to a public repository. For example, in addition tosummary statistics, the data points behind means, medians and variance measures should be available. If there are restrictions on publicly sharing data—e.g. participant privacy or use of data from athird party—those must be specified.

Reviewer #1: Yes

Reviewer #2: Yes

RESPONSE

Thank you.

COMMENT 4

4. Is the manuscript presented in an intelligible fashion and written in standard English?PLOS ONE does not copyedit accepted manuscripts, so the language in submitted articles must be clear, correct, and unambiguous. Any typographical or grammatical errors should be corrected atrevision, so please note any specific errors here.

Reviewer #1: Yes

Reviewer #2: Yes

RESPONSE

Thank you.

COMMENT 5 (Reviewer #1)

5. Review Comments to the Author

Reviewer #1: This large database is very well-written and give some interesting data about the relationship of various cancers with hypothyroidism. Only few minor suggestions had been advised.

RESPONSE

We sincerely appreciate this opinion of the reviewer.

COMMENT 5.1

1.Abstract - 'scarce influence' in the part of results should be changed into the another wordings

RESPONSE

According to this suggestion of the reviewer we have changed the expression 'scarce influence' to 'limited influence' in the Abstract of the revised version of the manuscript.

COMMENT 5.2

2.Intro - citation of estimated at 8.8% should be cited with standard format of reference

RESPONSE

Thanks for detecting this mistake. The citation from “Santos Palacios et al., 2018” was not included in our list of references by mistake.

In the new version of the manuscript, this reference is number 3, so we have had to modify the numbering of the rest of the references.

Santos Palacios S, Llavero Valero M, Brugos-Larumbe A, Díez JJ, Guillén-Grima F, Galofré JC. Prevalence of thyroid dysfunction in a Large Southern European Population. Analysis of modulatory factors. The APNA study. Clin Endocrinol (Oxf). 2018 Sep;89(3):367-375. doi: 10.1111/cen.13764.

COMMENT 5.3

3.Intro - first paragraph 'thyroid hypofunction' should be replaced with 'hypothyroidism'

RESPONSE

We have made this change based on the reviewer's suggestion. Please, see the new version of Introduction.

It is well known that hypothyroidism is frequency associated…

COMMENT 5.4

4.Page 9 - 'scarce influence' in the part of results should be changed into the another wordings

RESPONSE

In the Results section, we have replaced 'scarce influence' with 'limited influence', using the same expression as in the Abstract.

COMMENT 5.5

5.Page 10 - 3rd paragraph cite the reason why only less than 50% of patients diagnosed with hypothyroidism had been treated with LT4

RESPONSE

Thank you for this observation. We really cannot know the criteria that doctors use to decide on thyroid hormone replacement therapy in patients with hypothyroidism. The BDCAP database does not allow this information to be known. Therefore, we can only speculate on the reasons why more than half of the hypothyroid subjects did not receive thyroid hormone. Following the reviewer's recommendation, we have highlighted this fact in the new version of the manuscript.

In the Results section, we emphasize that the 42.84% figure is derived from the numerical information provided by the BDCAP.

Of the 2,282,124 patients diagnosed with hypothyroidism in 2018, BDCAP shows that there were 977,761 (42.84%) subjects with thyroid hormone replacement therapy (Table 4).

In the Discussion section we also clearly note that less than half of hypothyroid people received replacement therapy and that the BDCAP does not allow us to know the reasons for using or not using replacement therapy.

It should be noted that less than half of the hypothyroid patients were on replacement therapy. The BDCAP database does not allow knowing the criteria for treatment with thyroid hormone, so it is conceivable that a large part of untreated hypothyroid subjects presents only mild or transient subclinical hypothyroidism without the need for treatment.

COMMENT 5.6

6.Discussions were too long and should be more concise especially page 13-15

RESPONSE

We have revised the discussion text and shortened it noticeably, following this reviewer's recommendation

We have only expanded the part of the text that refers to bladder cancer, to address reviewer #1's comment 5.9 (see below).

As a consequence of this reduction of the text, we have omitted the following references: 24. Vorherr, 1978; 26. Ditsch et al., 2010; 45. Boelaert et al 2006.

COMMENT 5.7

7.Ref 20 refers to subclinical hypothyroidism only

RESPONSE

The reviewer's comment is correct. The meta-analysis by Gómez-Izquierdo et al (BMC Endocr Disord 2020;20:83) refers only to subclinical hypothyroidism. We have added the expression subclinical hypothyroidism in the paragraph referring to this study to make it more appropriate.

See the Discussion section in the new version.

Furthermore, a recent systematic review of controlled clinical trials and observational studies found no association between subclinical hypothyroidism and breast and prostate cancers

COMMENT 5.8

8.Ref 26 did not congruent with the written text

RESPONSE

Certainly, this citation is unfortunate, since the content of the article by Ditsch et al (2010) is not consistent with what is stated in our text. To avoid confusion for the reader, we have omitted this reference in the revised version of our manuscript.

COMMENT 5.9

9.Postulated mechanism why CA bladder did not associate with hypothyroidism should be discussed.

RESPONSE

Thank you for this comment. There is little information in the recent literature on the association between bladder cancer and hypothyroidism. There is also not much detailed information on the patho-physiological mechanisms that can explain the relationship between thyroid hormones or their deficiency and the genesis of bladder cancer. We have again reviewed the literature on this topic and have added a short paragraph to the discussion of the revised version of the manuscript, as suggested by the reviewer. In this new paragraph we have added a new bibliographical reference.

Subgroup analysis showed that bladder cancer had a peculiar behavior that was difficult to explain. The relative risk was increased in non-active persons and inhabitants of municipalities with 50,001 to 100,000 inhabitants, but reduced in active persons, pensioners and inhabitants of municipalities with 100,001 to 500,000 inhabitants. Our result contrasts with that reported by Mellemgaard et al [37] in a cohort of individuals who were discharged from a Danish hospital. These authors found an increased risk for bladder cancer among women. Differences may be accounted for by the fact that admitted patients may differ from the general population registered in BDCAP. Recently it has been shown that thyroid receptor-interacting protein 13 (TRIP13), a protein associated with the progression of several cancers, promotes proliferation and invasion of bladder cancer [39]. It can be speculated that in certain population groups, thyroid hormone deficiency might act by modifying the effect of certain proteins or carcinogen factors on the genesis of bladder neoplasms.

COMMENT 5 (Reviewer #2)

Reviewer #2: The study, based on a large sample, found that hypothyroidism increases the risk of a variety of malignancies. However, there are still two things that can be considered. First, what are the age groups based on. Second, whether there was a difference in the degree of hypothyroidism between the alternative treatment group and the non-alternative treatment group.

RESPONSE

Thank you for this comment. The first question refers to the age groups into which we have divided the Spanish population. Our age groups are based on the groups directly offered by the BDCAP database in its statistical portal (https://pestadistico.inteligenciadegestion.sanidad.gob.es/publicoSNS/S/base-de-datos-de-clinicos-de-atencion-primaria-bdcap).

The age of the subjects is one of the dimensions that allows us to study the "health problems" cube used in our study. The age dimension is divided into large groups that include children (0-14 years), young people (15-34 years), adults (35-64 years) and older people (65 and over).

We have slightly modified our description of these groups in the Material and Methods section.

age (large groups registered in BDCAP, that is, 0-14, 15-34, 35-64 and 65 years and over), …

Second, whether there was a difference in the degree of hypothyroidism between the alternative treatment group and the non-alternative treatment group.

The second observation question refers to the difference in the degree of hypothyroidism between the alternative treatment group and the non-alternative treatment group.

Unfortunately, we do not have this information as the BDCAP database does not distinguish between different degrees of hypothyroidism.

Naturally, we acknowledge this limitation of our study in the limitations paragraph of our Discussion.

We were also unable to know the etiology, duration or degree of hypothyroidism in our patients, since this information is not found in the BDCAP.

---

## [Editor Report · Decision Letter 1]

21 Sep 2022

Malignant neoplasms in people with hypothyroidism in Spain: a population-based analysis

PONE-D-22-16535R1

Dear Dr. Diez,

We’re pleased to inform you that your manuscript has been judged scientifically suitable for publication and will be formally accepted for publication once it meets all outstanding technical requirements.

Kind regards,

Donovan Anthony McGrowder, PhD., MA., MSc

Academic Editor

PLOS ONE

Additional Editor Comments):

Dear Dr. Diez, <o:p></o:p>

The manuscript was revised in accordance with the reviewers’ comments and is provisionally accepted pending final checks for formatting and technical requirements.

Regards,

Dr. Donovan McGrowder (Academic Editor)<o:p></o:p>

---

## [Editor Report · Acceptance letter]

26 Sep 2022

PONE-D-22-16535R1 

Malignant neoplasms in people with hypothyroidism in Spain: a population-based analysis 

Dear Dr. Díez:

I'm pleased to inform you that your manuscript has been deemed suitable for publication in PLOS ONE. Congratulations! Your manuscript is now with our production department. 

Kind regards, 

on behalf of

Dr. Donovan Anthony McGrowder 

Academic Editor

PLOS ONE